# Human Natural Antibodies Recognizing Glycan Galβ1-3GlcNAc (Le^C^)

**DOI:** 10.3390/ijms21186511

**Published:** 2020-09-05

**Authors:** Kira Dobrochaeva, Nailya Khasbiullina, Nadezhda Shilova, Nadezhda Antipova, Polina Obukhova, Oxana Galanina, Mikhail Gorbach, Inna Popova, Sergey Khaidukov, Natalia Grishchenko, Nikolai Tupitsyn, Jacques Le Pendu, Nicolai Bovin

**Affiliations:** 1Shemyakin-Ovchinnikov Institute of Bioorganic Chemistry, Russian Academy of Sciences, 16/10 Miklukho-Maklaya, 117997 Moscow, Russia; kira.dobrochaeva@gmail.com (K.D.); pumatnv@gmail.com (N.S.); antipova.nadine@gmail.com (N.A.); polina@carb.ibch.ru (P.O.); galox@inbox.ru (O.G.); gorbach-m-m@yandex.ru (M.G.); innussik.popova@gmail.com (I.P.); khsergey54@mail.ru (S.K.); 2Semiotik LLC, 16/10 Miklukho-Maklaya, 117997 Moscow, Russia; crosbreed@list.ru; 3National Medical Research Center for Obstetrics, Gynecology and Perinatology named after Academician V.I. Kulakov of the Ministry of Healthcare of Russian Federation, 4 Oparin str., 117997 Moscow, Russia; 4N.D. Zelinsky Institute of Organic Chemistry, Russian Academy of Sciences, 47 Leninsky pr., 119991 Moscow, Russia; 5Peoples’Friendship University of Russia (RUDN), 6 Miklukho-Maklaya, 117198 Moscow, Russia; 6National Research University Higher School of Economics, 101000 Moscow, Russia; 7N.N. Blokhin Russian Cancer Research Centre of the Ministry of Healthcare of Russian Federation, 24 Kashyrskoe Sh., 115478 Moscow, Russia; natali-2712@mail.ru (N.G.); nntca@yahoo.com (N.T.); 8CRCINA, INSERM, Université d’Angers, Université de Nantes, 8 quai Moncousu, BP 70721-44007 Nantes, France; Jacques.Le-Pendu@univ-nantes.fr; 9Centre for Kode Technology Innovation, Auckland University of Technology, 55 Wellesley Street East, Auckland 1010, New Zealand

**Keywords:** breast cancer, cancer-associated antibodies, Le^C^ antigen, natural anti-glycan antibodies, printed glycan array

## Abstract

The level of human natural antibodies of immunoglobulin M isotype against Le^C^ in patients with breast cancer is lower than in healthy women. The epitope specificity of these antibodies has been characterized using a printed glycan array and enzyme-linked immunosorbent assay (ELISA), the antibodies being isolated from donors’ blood using Le^C^-Sepharose (Le^C^ is Galβ1-3GlcNAcβ). The isolated antibodies recognize the disaccharide but do not bind to glycans terminated with Le^C^, which implies the impossibility of binding to regular glycoproteins of non-malignant cells. The avidity (as dissociation constant value) of antibodies probed with a multivalent disaccharide is 10^−9^ M; the nanomolar level indicates that the concentration is sufficient for physiological binding to the cognate antigen. Testing of several breast cancer cell lines showed the strongest binding to ZR 75-1. Interestingly, only 7% of the cells were positive in a monolayer with a low density, increasing up to 96% at highest density. The enhanced interaction (instead of the expected inhibition) of antibodies with ZR 75-1 cells in the presence of Galβ1-3GlcNAcβ disaccharide, indicates that the target epitope of anti-Le^C^ antibodies is a molecular pattern with a carbohydrate constituent rather than a glycan.

## 1. Introduction

Natural antibodies (nAbs) capable of binding to Galβ1-3GlcNAcβ disaccharide (Le^C^) have been identified in the blood of more than 95% of healthy donors [1,2,3]; their typical titers are much higher than, for example, the *allo* antibody titers against blood group A or B antigens or xenoantibodies against the alpha-Gal epitope [2]. The antibodies (Abs) have an intriguing epitope specificity; they bind the disaccharide and oligosaccharides of the general structure of hexose1-3Galβ1-3GlcNAcβ1-O-sp (sp, spacer group) but are incapable of binding Galβ1-3GlcNAcβ1-3Galβ1-4Glc and other glycans of cellular glycoproteins carrying the disaccharide Le^C^ as a terminal fragment of the carbohydrate chain [4]. This specificity explains why antibodies with a high blood level (~5 µg/mL) do not cause an autoimmune reaction against Le^C^-terminated cell surface glycoproteins. There are a number of data that make us consider anti-Le^C^ nAbs to be involved in anti-cancer surveillance. First, their titers in patients with breast cancer are significantly lower than in healthy people [4]. Second, isolated human anti-Le^C^ nAbs stain breast cancer tissue [5]. Third, these antibodies bind B cells in tumor lesion milieu [5]. Fourth, in studies aimed at finding diagnostic signatures (a signature usually consists of 6-10 anti-glycan nAbs), these antibodies turned out to be the most frequent constituent of the signature [6,7,8]. In addition, two monoclonal antibodies with similar specificities are known—LU-BCRU-G7, which specifically binds to breast cancer tissue [9] and 58-1, which was generated using CA19.9 glycoprotein as an immunogen [10] (Specificity and comparison of monoclonal antibodies (mAb) with human anti-Le^C^ are presented in Reference [10]). Taking into account all the above data, here we aimed at (1) characterizing in more detail the epitope specificity of human anti-Le^C^ with newly synthesized glycans, in order to determine which glycan could be the target molecule for anti-Le^C^ antibodies in vivo; (2) finding target cells or tissues to which the anti-Le^C^ antibodies bind; and (3) comparing human and mouse nAbs against Le^C^ and answering the question of whether a mouse model can be used to study in vivo the processes triggered by the antibodies.

## 2. Results

### 2.1. Epitope Specificity of Mouse Anti-Le^C^ Antibodies

Since the epitope specificity of human anti-Le^C^ nAbs appear to be unusual, human and mouse antibodies were compared. The antibodies were isolated under the same conditions with the same adsorbent as human antibodies [1,4]. Because the quantity of mouse serum is limited, we had to measure the sum of immunoglobulin G + M (IgG + IgM) antibodies. As a source of the antibodies, pooled mouse sera were used. The printed glycan array (PGA) analysis data are presented in Appendix A; the 15 top ligands are shown in Table 1. 

We can indirectly assume that similarly to human antibodies, the affinity of mouse anti-Le^C^ for Le^C^ disaccharide is moderate, because this glycan occupies not the top position (Table 1), moreover, the difference (measured in relative fluorescence units, RFU) between Le^C^ disaccharide and the top glycans is as high as an order of magnitude. The characteristic feature of mouse anti-Le^C^ is the ability to bind the disaccharide epitope in the terminal or pendant position (underlined in Table 1) of all top glycans and the inability to recognize it at the innermost position, for example, in Galβ1-4GlcNAcβ1-3Galβ1-3GlcNAcβ (see Appendix A); that is, murine anti-Le^C^ antibodies resemble typical adaptive and natural anti-glycan Abs, such as anti-Galili (xeno-epitope) and anti-blood group A(B) immunoglobulins but not human anti-Le^C^.

### 2.2. Anti-Le^C^ antibodies in Infants Below 12 Months of Age

The level of anti-Le^C^ IgM antibodies was traced in 22 newborns (their sera were probed with PGA) until the age of 12 months. In contrast to almost all other anti-glycan antibodies, RFU values for anti-Le^C^ Abs in infants were much lower than in their mothers, healthy non-pregnant donors and in complex immunoglobulin preparation (CIP, immunoglobulin derived from >1000 donors, containing IgG+IgM+IgA). Table 2 shows the data on five infants (22 infants were divided in groups according to the type of food, data for one group is shown; this study is described in more detail in Reference [11]). Here, we consider the Le^C^ family of glycans, that is, glycans with obvious structural similarity to the Galβ1-3GlcNAc disaccharide that are capable of binding the affinity-isolated (see below) anti-Le^C^ Abs. Infants at the age of 12 months still have no antibodies to all glycans of this family (Table 2).

### 2.3. Adjusted Epitope Specificity of Human Anti-Le^C^ Antibodies

Human anti-Le^C^ nAbs were isolated as described previously using a two-step process [1,4], where the first step is affinity chromatography on the Le^C^-Sepharose and the second stage—by exhausting of the eluted material on ligand-free Sepharose. An IgM/IgG/IgA ratio of anti-Le^C^ antibodies isolated from CIP was 4:1:1, it was determined using an ELISA. For a more accurate than previous profiling of the specificity of antibodies [1], we used a PGA containing a number of newly synthesized Le^C^ analogs, for example, Galβ1-3GlcNAcβ1-3GalNAcα (Le^C^-Tn) [10], Galβ1-3GlcNAcα1-3Galβ1-3GlcNAcβ (Le^C^α3’Le^C^) and similar glycans (Appendix A) [12]. Since the content of IgG (isolated from CIP) against Le^C^ is low, here we focus on IgM antibodies (all IgG and IgM data are presented in Appendix A). Top-rank glycans are presented in Table 3.

The following features of the antibodies should be noted: (1) the elongation of the Galβ1-3GlcNAcβ inner core with a different substituent (Su, Sia, mono- or disaccharides or a hydrophobic benzyl group) does not increase the binding rate significantly; (2) we have confirmed earlier data that the antibodies do not bind molecules of the Galβ1-3GlcNAcβ-glycan type, with the exception of two low-affinity artificial ligands, Galβ1-3GlcNAcα1-3Galβ1-4GlcNAcβ and Galβ1-3GlcN(Fm)β1-3Galβ1-4GlcNAcβ, where Fm is the formyl group, –C(O)H, (the lowest in Table 3); (3) the Galα1-3GlcNAcβ and L-Fucβ1-3GlcNAcβ disaccharides have an affinity equal to that of the parent Galβ1-3GlcNAcβ disaccharide; (4) the unsubstituted GlcNAcβ monosaccharide retains some affinity, despite the absence of the Galβ moiety.

Similar results were obtained when pooled sera of healthy individuals (103 donors) were tested (Appendix A, data obtained using a Consortium for Functional Glycomics (CFG) chip containing several complex natural N-glycans with Le^C^ termination, http://functionalglycomics.com/glycomics/publicdata/selectedScreens.jsp, Glycan Array: 2505,) instead of CIP; data for all top glycans coincided, although their ranks were slightly different. In most experiments, we used an IgM-containing immunoglobulin preparation (CIP), which provided an averaged picture, because this type of Ig preparations is obtained from 1000 or more blood samples, which is unrealistic if serum samples are pooled. Nevertheless, we tested the pool (103 donors) of sera and obtained similar data for Abs isolated from the immunoglobulin preparation and from pooled serum. We also observed consistency between different CIP batches (data not shown).Note that the CFG array contains glycans with the common formula X-Galβ1-3GlcNAcβ1-6GalNAcα-, all glycans of this type displayed moderate to high RFU values. In contrast, the isomeric Galβ1-3GlcNAcβ1-3GalNAcα- structure (in our routine array, from Semiotik LLC) exhibited lack of binding (not included in Table 3).

### 2.4. GlcNAcβ-OSer Is Not a Target for Anti-Le^C^ Human nAbs

As seen from Table 3, the GlcNAcβ monosaccharide is only slightly inferior to the Le^C^ disaccharide in terms of the affinity for the antibodies; therefore, it could be assumed that anti-Le^C^ are “designed” to recognize a family of glycoproteins with O-glycosylation, such as GlcNAcβ-*O*-Ser/Thr, which is typical for intracellular glycoproteins. Inhibition ELISA showed that the difference in affinity for the Le^C^/GlcNAcβ pair is small; hence, the serine residue instead of the simple three-carbon spacer group significantly impairs affinity (Appendix A). Thus, the intracellular GlcNAcβ-*O*-Ser/Thr glycotope is unlikely to be the actual target for anti-Le^C^ nAbs.

### 2.5. Histochemistry

The labeling of human healthy tissues and colorectal carcinoma specimens with biotinylated human anti-Le^C^ antibodies was performed to determine the tissue specificity of these antibodies. Most healthy tissues studied were stained with anti-Le^C^ antibodies (Appendix A) but the staining was weak despite the high concentrations of antibodies used (three orders of magnitude higher than in the blood of healthy subjects). Only in the liver was an intense diffuse labeling of hepatocytes observed (Appendix A). In other organs, staining, if any, was confined to epithelial cells and was exclusively intracellular (Appendix A). No staining of the apical cell surface or secreted material was observed, probably, because the recognition of the precursor motif was masked upon further elongation of glycan chains or substitution with additional monosaccharides. In human colon carcinoma, diffuse intracellular labeling of carcinoma cells was more intense than that of the corresponding healthy epithelial cells (Appendix A). These differences were preserved upon a decrease in the concentration of the antibodies to their level in the donor blood. Additional strong staining of mononuclear infiltrating cells was also visible in some cases, both within the tumor and in the adjacent normal tissue (Appendix A).

### 2.6. Binding of Anti-Le^C^ nAbs to Cells, Flow Cytometry

We tested cultures of three breast cancer cell lines, HS578t, ZR 75-1 and SKBR-3, for Le^C^-positive cells (Figure 1). The former two lines were found positive and SKBR-3, negative. In subsequent experiments, we used ZR 75-1 cells, because HS578t cells were difficult to grow.

The staining increased with time and reached a plateau within 120 min of incubation (Figure 2). It should be emphasized that the binding strongly depended on the cell density (Figure 3). In the monolayer with a low density, only 7% of the cells were positive, whereas in the monolayer with the maximum density, their number was increased to 96%. Binding preferably occurs intracellularly, while surface staining is less intense (data not shown).

### 2.7. Binding of Anti-Le^C^nAbs to ZR 75-1 Cells in the Presence of Galβ1-3GlcNAc Disaccharide

When binding of affinity-isolated anti-Le^C^ nAbs was performed at 37 °C in the presence of free (unconjugated) Le^C^ in the form of Galβ1-3GlcNAcβ1-OCH_2_CH_2_CH_2_NH_2_ disaccharide, unexpectedly, an increase in the number of positive cells was observed instead of a decrease (Appendix A). In previous publication [1] and the article in press [13] we demonstrated only minor effect of the nature of the spacer on the binding of antibodies in ELISA and PGA, so here we took only a disaccharide with a C3 spacer. The effect of enhancement with Galβ1-3GlcNAcβ1-OCH_2_CH_2_CH_2_NH_2_ was reproduced at 25 °C (data not shown). At either temperature, the effect is not sensitive to an increase in the disaccharide concentration from 9 to 90 mM.

## 3. Discussion

Anti-Le^C^ nAbs belong to the group of the most representative natural antibodies in human peripheral blood; the fact that they are found in almost all (~95%) people leaves no doubt that anti-Le^C^ antibodies are necessary for the functioning of innate immunity [2]. Their function appears to be species-specific, because we have found that mice have no nAbs resembling human anti-Le^C^ in epitope specificity. This has been demonstrated for antibodies affinity-isolated (with Le^C^-Sepharose) under the same conditions as the human nAbs. PGA analysis of this material has shown that we have obtained fundamentally different antibodies that lack the key characteristic of human nAbs, the capacity for recognizing preferentially the internal Galβ1-3GlcNAcβ motif. This dramatic difference probably means that the mouse antibodies do not have the anti-cancer function supposed for anti-Le^C^ in humans. It can be argued that this function is not required in the first 12 months of life, since young children do not have anti-Le^C^; moreover, the difference between the titers of anti-Le^C^ in children and adults is the largest among all anti-glycan nAbs studied. The affinity of human anti-Le^C^ antibodies for monomeric and multimeric (polymer carrier-bound) disaccharides has been estimated; in the former case, K_d_ is equal to 10^−4^ M (Appendix A) and in the latter case, K_d_ is 10^-9^ M (Appendix A). The nanomolar level of K_d_ for the multivalent (IgM) interaction corresponds to the antibody concentration of anti-Le^C^ in the blood (~5 nM) and indicates that the blood concentration of the antibodies is sufficient for the physiological interaction of their cognate antigen.

Taking into account the unique properties of antibodies against Le^C^ mentioned above, we considered it necessary to investigate in more detail their epitope specificity and try to figure out their actual molecular target. The most intriguing aspect of epitope specificity of natural anti-Le^C^ antibodies is the role of aglycon radical—the Galβ1-3GlcNAcβ-R motif can be recognized only if R = OH or is not a bulky group, such as a small spacer. The Galβ1-3GlcNAcα1-3Galβ1-4GlcNAcβ tetrasaccharide is the only one in the Le^C^-LacNAc series that displays any affinity for the antibodies but the glycosidic bond between two disaccharides is not natural. The carbohydrate chain Galβ1-3GlcNAcβ1-3Galβ1-4GlcNAc is rather typical of human glycoproteins, while its *gluco*-analog, that is, Galβ1-3GlcNAcβ1-3Galβ1-4Glc, is one of the main components of breast milk. Using the PGA version 5.0 of the CFG, which has (unlike PGA which we routinely use in our laboratory) bi- and triantennary *N*-chains with Le^C^ ends, as shown below, we have confirmed the absence of any interaction of the isolated nAbs with the Le^C^ motif in the composition of common *N*-chains (Figure 4A).

Does this mean that there are no other options for disposition of Le^C^ as a terminal motif in natural glycoconjugates? The biosynthetic machinery of malignant cells produces rare, untypical chains based on the Galβ1-3GlcNAc backbone, such as the type 1 glycans Neu5Acα2-3Galβ1-3GlcNAc (in the CA-50 tumor marker) and Neu5Acα2-3(Fucα1-4)Galβ1-3GlcNAc (in the CA19.9 tumor marker) [14]. The elevated synthesis of CA-50 and CA19.9 is explained by a higher β-galactosyltransferase activity in the tumor cell, which, in turn, implies the ability of this enzyme to synthesize unknown tumor-associated glycans, in whose structure the Galβ1-3GlcNAc motif attached to a structure other than Galβ1-4GlcNAc. Here, it is necessary to cite data on the production of mouse Abs against the CA19.9 cancer glycoprotein—two hybridomas generated in that study resembled human natural Abs against Le^C^ in terms of the specificity of IgM antibodies [10]. Because the biosynthetic machinery does not exclude the emergence of variants with the Galβ1-3GlcNAc disaccharide attached to the *O*-chains (Figure 4B–C), that is, to GalNAcα-Ser/Thr, we chemically synthesized the Galβ1-3GlcNAcβ1-3GalNAcα trisaccharide and added it to our PGA. The results of the analysis using this PGA were weakly positive. In contrast, the affinity of anti-Le^C^ nAb for Galβ1-3GlcNAcβ1-6GalNAcα (a 1-6 isomeric trisaccharide, Figure 4C) was close to that of glycans from the above-mentioned Le^C^ family—it turned out that this is the only structural motif potentially found in nature (in glycoprotein *O*-chains) capable of binding anti-Le^C^ Abs. Therefore, we cannot exclude that Galβ1-3GlcNAcβ1-6GalNAcα or its elongated form X-Galβ1-3GlcNAcβ1-6GalNAcα (X is Neu5Ac or an oligosaccharide) is the actual target for cancer-associated anti-Le^C^ antibodies or a key constituent of a more sophisticated antigen (see below).

In this study, we have made significant progress in understanding the nature of the carbohydrate epitope recognized by anti-Le^C^ antibodies, which we isolate from donor blood using Le^C^-Sepharose but what we have come to is puzzling. Below, we summarize the results that do not fit well within the framework of usual ideas about the interaction of antibodies with carbohydrate antigens. (1) The Abs do not bind the Galβ1-3GlcNAcβ-OR motif in the molecules of real natural glycans, except when R = −6GalNAcα, a structure that has not been found yet but that could be synthesized by known glycosyltransferases. (2) For antibodies isolated using Le^C^-Sepharose, the Le^C^ disaccharide itself is not the top ligand (Table 3); several glycans that have not been found in humans (and, moreover, whose appearance is not allowed biochemically), such as Galα1-3GlcNAc and Le^C^α1-3’Le^C^, have a higher affinity. (3) The contribution of the Galβ fragment to the interaction with antibodies is insignificant. We have found that *hexose*-3GlcNAcβ-OR (where *hexose* is not Galβ) can bind antibodies equally strongly; that is, GlcNAcβ- is the key motif of the epitope. (4) The AcNH group of GlcNAcβ is beneficial but not critical; the same is true for the 3^OH^ group of GlcNAcβ. Summing up, from the data presented a paradoxical conclusion could be drawn that 4^OH^ and 5^CH2OH^ of GlcNAc form a key epitope. However, the data on the absence (or weak) binding of the antibodies to many of other PGA glycans containing the same combination of 4^OH^ and ^5CH2OH^ groups suggest the opposite. This again points to the unusual “architecture” of the anti-Le^C^ antibody epitope.

We consider two hypotheses to explain these unexpected results. First, anti-Le^C^ nAbs may bind a spatial, conformational epitope, like, for example, 2G12 monoclonal antibodies capable of recognizing a supramolecular epitope formed by two closely located mannose-reach *N*-chains in the HIV glycoprotein [15],or human antibodies capable of binding the GM1/GD1a ganglioside complex [16], i.e., an epitope representing a molecular pattern rather than classical linear structure. Second, the antibodies may bind a peptide epitope rather than a carbohydrate one; that is, the Le^C^ disaccharide may be a mimetic for an unknown protein target. Anyway, we do not know the exact structure of the actual epitope, the more so as we deal with polyclonal Abs.

Therefore, we tried to approach the problem from another direction, namely, to find a cellular or tissue target of anti-Le^C^ nAb and, knowing it, look for the molecular target. For this purpose, we proceeded from the assumption that the functional target of anti-Le^C^ Abs is a tumor cell. Because secondary Abs against human immunoglobulins can cross-react with human tissues, we used biotinylated human anti-Le^C^. The labeled Abs stained some specimens in a tissue microchip, preferably of epithelial origin. The most important result of this pilot experiment is that the binding mainly occurred inside the cell rather than on the membrane, similarly to the interaction of the Abs with ZR 75-1 cells (see below).

More definite results were obtained using cell lines derived from human breast tumors. This choice was determined by previous data demonstrating the binding of anti-Le^C^ to breast cancer tissue [5]. Of the three tested cell lines, HS578T and ZR 75-1 were Le^C^-positive, while SKBR-3 was Le^C^-negative. It is known that HS578T cells have been derived from mammary gland/breast carcinoma cells; ZR 75-1 cells have been derived from a metastatic site of ductal carcinoma; this cell line is characterized by high levels of expression of MUC-1 and, to a lesser degree, MUC-2 glycoproteins but not MUC-3 glycoprotein [17]; SKBR-3 cells have been derived from metastatic adenocarcinoma (https://www.lgcstandards-atcc.org/). Due to some technical difficulties of handling HS578T cells, we subsequently used only ZR cells.

Up to 96% of ZR 75-1 in a monolayer were found to be Le^C^-positive. Their interaction with the nAbs had three unusual features. First, in the case of permeabilized cells, both intracellular and cell-surface stainings were observed; this agreed with tissue microarray data. We suppose that anti-Le^C^ IgM antibodies cause cells damage (microscopy reveals a slightly changed morphology), which is the reason for the intracellular staining. Because most of intracellular proteins are O-glycosylated with GlcNAcβ monosaccharide (or a more complex hexose-GlcNAcβ residue) ([18] and anti-Le^C^ nAbs are capable of binding GlcNAcβ (see above), we examined the possibility that the intracellular binding is mediated by O-GlcNAc-*ated* proteins. However, inhibition ELISA demonstrated the absence of inhibition by free GlcNAcβ-*O*Ser, which agrees with the fine epitope specificity of anti-Le^C^ Abs (see discussion above).

The second unusual feature of the interaction of anti-Le^C^ nAbs with ZR 75-1 cells is a dramatic dependence of staining on cell density—only a slight staining occurred in a ~50% cell monolayer, while up to 96% of cells were positive at the maximum density. It is yet unclear whether this effect is the result of newly formed contacts between cells or the manifestation of another mechanism.

The third unusual feature of the interaction of anti-Le^C^ nAb with ZR 75-1 cells is an increased staining in the presence of free Galβ1-3GlcNAc disaccharide instead of the expected inhibition. In contrast, in an “artificial” ELISA, where the immunological plate is coated with polyacrylamide glycocojugate Galβ1-3GlcNAc-PAA, free unconjugated disaccharide Galβ1-3GlcNAc demonstrates the classical effect of dose-dependent inhibition at concentration comparable with that used in cell analysis (see Appendix A). Apparently, the human anti-Le^C^ antibody paratope is larger than that of adaptive antibodies; one of its regions recognizes an unknown epitope and another one, the epitope that we here call Le^C^. This model explains the observed paradoxically increased binding instead of inhibition when the antibodies interact with cells—as a result of the “doping” with the Galβ1-3GlcNAc disaccharide, the paratope becomes less flexible and, therefore, binds the epitope of ZR 75-1 cells better. The phenomenon of polyreactivity is well known for nAbs. The reason why the paratope of anti-glycan nAbs may have two binding regions is discussed in the work of Bovin et al., 2020 (in press) [13]. The accumulated data on the specificity of anti-Le^C^ antibodies suggest the recognition of a spatial, composite epitope, the exact architecture of which can hardly be determined by existing methods.

## 4. Materials and Methods 

### 4.1. Reagents

Unlabeled rabbit anti-human Ig(G+M+A), anti-human IgG-biotin, anti-human IgM-biotin and Str-HRPO (streptavidin-horseradish peroxidase) were obtained from Southern Biotechnology Associates, Inc. (Birmingham, AL, USA). Goat anti-human-IgG labeled with Alexa^555^, goat anti-human-IgM labeled with Alexa^647^ and streptavidin labeled with Alexa^555^ (Str-Alexa^555^) were from Invitrogen (Eugene, OR, USA). Biotinylated goat anti-mouse Ig(G+M+A) were obtained from Thermo Fisher Scientific (Rockford, MN, USA). Tween-20, human IgG and IgM, phosphate buffered saline (PBS) and glycine were obtained from Sigma (St. Louis, MO, USA). Tris and bovine serum albumin (BSA, fraction V) were from Serva (Heidelberg, Germany). Salts and acids were of analytical grade, they were from Honeywell Int. (Mexico City, Mexico) or Merck (Kenilworth, NJ, USA). The AEC (3-amino-9-ethylcarbazole) detection kit was from Vector Laboratories (Marion, IA, USA). MaxiSorp 96-well microtiter immunoplates (hereinafter, plates) were obtained from Thermo Fisher Scientific (Rochester, MN, USA). PAA-conjugated glycans (30 kDa), spacer-armed glycans and affinity glycoadsorbents for isolation of antibodies were from GlycoNZ (Auckland, New Zealand). Glycochips were produced by Semiotik LLC (Moscow, Russia). Complex immunoglobulin preparation (CIP) was from Microgen (Moscow, Russia). The Biotinylation Kit was from Sileks GmbH (Badenweiler, Germany). Human cell lines ZR 75-1 and HS578t were purchased from the Institute of Cytology of the Russian Academy of Science (St. Petersburg, Russia); SKBR-3 cells were purchased from ATCC (Manassas, VA, USA). DMEM (Dulbecco’s modified Eagle’s medium), L-glutamine, supplemented with 10% fetal calf serum was from Paneko (Moscow, Russia). Serum-free Protein Block solution was from Dako (Carpinteria, CA, USA).

### 4.2. Isolation of Human anti-Le^C^ Antibodies

CIP (35 mg/mL in PBS/0.3% BSA, 300 mg per 1 mL of adsorbent) was applied (15 cm/h) to a column containing Galβ1-3GlcNAcβ-PAA-Sepharose 6FF (referred to here as Le^C^-Sepharose). The column was pre-washed and equilibrated with PBS. Then, the column was washed with PBS (60 cm/h) to the baseline optical density (OD,280 nm) value. The bound anti-Le^C^ antibodies were eluted with 0.2 M TrisOH/0.5 M NaCl (pH 10.2, 15 cm/h) with immediate subsequent neutralization of the eluate with 2 M glycine-HCl (pH 2.5) to a final pH of 7.3–7.5. The antibodies were dialyzed against PBS and concentrated by centrifugation in an Amicon Ultra Centrifugal Filter Unit, 100 kDa cutoff (Merck, Darmstadt, Germany) at 4 °C, 3000 g. Then, anti-Le^C^ antibodies (0.5–3.0 mg/mL in PBS, 10 mg per 1 mL of adsorbent) were applied (30 cm/h) to a mini-column containing PAA-Sepharose 6FF, the column was pre-washed and equilibrated with PBS. All unbound material (the flow-through fraction containing the target anti-Le^C^ antibodies) was collected and concentrated by centrifugation using an Amicon Ultra Centrifugal Filter Unit, 100 kDa cutoff (Merck, Darmstadt, Germany) at 4 °C, 3000 g. Fractions were monitored using a UV detector (Bio Rad, Philadelphia, PA, USA) at 280 nm. All the buffers used contained 0.02% NaN_3_.

### 4.3. Isolation of Mouse Antibodies Using the Le^C^-Sepharose Affinity Adsorbent

Sera from 23 αGalT knock-out mice was pooled, the total protein concentration was 57 mg/mL. The pooled serum was diluted with PBS to a total protein concentration of 35 mg/mL and was applied (280 mg per 1 mL of adsorbent; 75 cm/hr) to a column containing Galα1-3Galβ1-4GlcNAcβ-PAA-Sepharose 6FF (to isolate anti-αGal antibodies necessary for an unrelated study). The column was pre-washed and equilibrated with PBS. The flow-through fraction was collected and applied (240 mg per 1 mL of adsorbent; 75 cm/hr) to a pre-washed and PBS-equilibrated column containing Le^C^-Sepharose. Then, the column was washed with PBS (60 cm/h) to the baseline OD (280 nm) value. The bound anti-Le^C^ antibodies were eluted with 0.2 M TrisOH/0.5 M NaCl (pH 10.2, 15 cm/h), which was followed by immediate neutralization of the eluate with 2 M glycine-HCl (pH 2.5) to final a pH of 7.3–7.5. The antibodies were dialyzed against PBS and concentrated by centrifugation using an Amicon Ultra Centrifugal Filter Unit, 100 kDa cutoff (Merck, Darmstadt, Germany) at 4 °C, 3000 g. Fractions were monitored using a UV detector (Bio Rad, Philadelphia, PA, USA) at 280 nm. All the buffers used contained 0.02% NaN_3_.

### 4.4. Biotinylation of Antibodies

The isolated and purified anti-Le^C^ antibodies were labeled as recommended by the manufacturer of the kit (Sileks GmbH, Germany) for biotinylation of proteins. Briefly, 0.5 mL of a conjugation buffer and 50 µL of a freshly prepared solution of the reagent for biotinylation (10 mg/mL) were added to 0.5 mL of anti-Le^C^ antibody solution (2 mg/mL in PBS, pH 7.4). The reaction mixture was incubated for 2 h at 22 °C. The obtained conjugate was dialyzed against PBS and concentrated by centrifugation in an Amicon Ultra Centrifugal Filter Unit, 100 kDa cutoff (Merck, Darmstadt, Germany) at 4 °C, 3000*g.*

### 4.5. Quantitation of Human Immunoglobulins

Plates were coated with rabbit anti-human Ig(G+M+A), 5 µg/mL (100 µL per well) in 0.05 M sodium carbonate buffer, pH 9.6, for 1 h at 37 °C, blocked with 3% BSA in PBS for 1 h at 37 °C and washed three times with the PBST-0.1%washing buffer (0.1% Tween-20 in PBS). Serial dilutions of human IgM, IgG and IgA (controls for the calibration curve plotting) or samples to be tested were added. The plates were incubated for 1 h at 37 °C and washed three times with PBST-0.1%. Then, the plates were incubated with biotinylated anti-human IgG (or IgM or IgA) antibodies (1:4000 in PBS containing 0.3% BSA) for 1 h at 37 °C and washed. The plates were incubated with Str-HRPO conjugate (1:2000 in PBS containing 0.3% BSA) and washed. Finally, the color was developed by a 30-min incubation with 0.1 M sodium phosphate/0.1 M citric acid buffer containing 0.04% *o*-phenylenediamine and 0.03% H_2_O_2_ and the reaction was stopped by adding 50 µL of 1 M H_2_SO_4_. The absorbance at 492 nm was recorded in a Multiskan MCC/340 microtiter plate reader (Perkin Elmer, Turku, Finland). The control wells were done without immunoglobulins. All assays were carried out in at least three replicates.

### 4.6. Inhibition ELISA

Plates were coated with PAA-conjugated glycans [19], 10 µg/mL in PBS, pH 7.4, for 1 h at 37 °C, blocked with 3% BSA in PBS for 1 h at 37 °C and washed three times with the PBST-0.1% washing buffer. Serial dilutions of inhibitors (50 µL per well; initial concentration, 8 to 1 mM) were prepared and anti-Le^C^ antibodies (0.25 µg/mL in PBS containing 0.3% BSA; 50 µL per well) were added. The plates were incubated for 1 h at 37 °C, washed three times with the washing buffer, incubated with biotinylated anti-human IgM antibodies (1:4000 in PBS containing 0.3% BSA) and washed. The plates were incubated with Str-HRPO conjugate (1:2000 in PBS containing 0.3% BSA) and washed. Finally, the color was developed by a 30-min incubation with 0.1 M sodium phosphate/0.1 M citric acid buffer containing 0.04% *o*-phenylenediamine and 0.03% H_2_O_2_, the reaction was stopped by adding 50 µL of 1 M H_2_SO_4_. The absorbance at 492 nm was recorded. The degree of inhibition was calculated in percent as (OD_A_ − OD_I_) × 100/OD_A_, where OD_A_ is the mean optical density in the absence of the inhibitor and OD_I_ is the mean optical density in the presence of the inhibitor.

### 4.7. Flow Cytometry

The ZR 75-1 and HS578t human cell lines were maintained in DMEM, L-glutamine, supplemented with 10% fetal calf serum. For immunofluorescent staining, a pellet containing 1 × 10^6^ cells (200 µL) was mixed with human anti-Le^C^ antibodies (5 µg/mL in PBS, 25 μL) and incubated at 37 °C for 40 min (or 20, 40, 60 and 120 min for time-dependent staining). The cells were washed three times for 10 min in PBST-0.1% containing 0.1% BSA and precipitated by centrifugation at 320*g* for 5 min. Then, the cells were incubated with goat anti-human IgM (µ-chain) conjugated with Alexa^647^, diluted 1:1000 in PBS containing 0.1% BSA. After that, the cells were washed three times for 10 min in PBST-0.1%with 0.1% BSA and resuspended in PBS for flow cytometry analysis. The stained cells were analyzed using a Cytomics FC 500 flow cytometer (Beckman Coulter, Inc., Miami, FL, USA). The fluorescence was measured and expressed as the mean fluorescence intensity (MFI). The cells treated only with the secondary mAbs were used as a negative control.

### 4.8. Flow Cytometry. Inhibition Version

ZR 75-1 cells were maintained in DMEM, L-glutamine supplemented with 10% fetal calf serum. Human anti-Le^C^ antibodies were incubated with PBS (control) containing 9 or 90 mM Galβ1-3GlcNAcβ-OCH_2_CH_2_CH_2_NH_2_ at 37 °C for 30 min. The next steps of immunofluorescent staining were carried out as described above.

### 4.9. Flow Cytometry Protocol for Intracellular Staining

ZR 75-1 human cells were maintained in DMEM, L-glutamine, supplemented with 10% fetal calf serum. The cells were fixed with 4% paraformaldehyde at room temperature for 10–15 min and permeabilized with 0.2% saponin (together with serum-free Protein Block solution for blocking) at 22 °C for 1 h. For immunofluorescent staining, a pellet containing 1 × 10^6^ cells (200 μL) was mixed with the solution of human anti-Le^C^ antibodies (5 µg/mL in PBS, 25 μL) and incubated at 37 °C for 40 min. The cells were washed three times for 10 min in PBST-0.1% containing 0.1% BSA and precipitated by centrifugation at 320*g* for 5 min. Then, the cells were incubated with goat anti-human IgM (µ-chain) conjugated with Alexa^647^, diluted 1:1000 in PBS containing 0.1% BSA. After that, the cells were washed three times for 10 min in PBST-0.1% with 0.1% BSA and resuspended in PBS for flow cytometry analysis. The stained cells were analyzed using a Cytomics FC 500 flow cytometer (Beckman Coulter, Inc., Miami, FL, USA). The fluorescence was measured and expressed as MFI. The cells treated only with the secondary mAbs were used as a negative control.

### 4.10. PGA Assay

A microarray on glass microscope slides (Semiotik LLC, Moscow, Russia, hereinafter referred as a chip) contains more than 400 synthetic glycans (>95% purity). The complete list of oligosaccharides is presented in Appendix A. The glycan concentration was 50 µM. All ligands were printed at six replicates onto NHS-activated Slide H (Schott Nexterion, Jena, Germany) as described in [20] by means of a sciFLEXARRAYER S5 non-contact piezo-arrayer (Scienion, Berlin, Germany). The drop volume was approximately 0.9 nL. Isolated antibodies (10 µg/mL) in PBST-0.1% and 1% of BSA were applied unto the chip (pretreated with PBST-0.1% for 15 min) and incubated in a humidified chamber at 37 °C for 1 h with mild rotation. The chip was washed with PBST-0.05% (0.05% Tween 20 in PBS)and incubated in a humidified chamber at 37 °C for 1 h (with mild rotation) in the presence of goat anti-mouse IgG+IgM+IgA (H+L) antibodies conjugated with biotin diluted 1:100 in PBST-0.1% containing 1% BSA in the case of mouse antibodies or with a mixture of goat anti-human IgM (µ-chain) conjugated with Alexa^647^ and goat anti-human IgG(H+L) conjugated with Alexa^555^ diluted 1:200 in PBST-0.1% containing 1% BSA in the case of human antibodies. After washing with PBST-0.05%, the chip with isolated human antibodies was finally washed with MilliQ water and dried by mild centrifugation. In the case of mouse antibodies, the chip was then incubated in the dark at 25 °C for 45 min (with mild rotation) with Str-Alexa^555^diluted 1:1000 in PBST-0.1%. Then, after washing with PBST-0.05% followed by MilliQ water rinsing, it was dried by mild centrifugation. All chips were scanned using an Innoscan 710 Fluorescence Scanner (Innopsys, Carbonne, France) at a resolution of 10 µm. The images were processed using a ScanArray Express 4.0 (with a fixed circle method) and, subsequently, the Microsoft Excel. Signals were measured as the median with interquartile deviation (Q1 and Q3 quartiles).

Profiling of human antibodies isolated from pooled sera was performed with a CFG chip, version 5.0 (http://functionalglycomics.com/glycomics/publicdata/selectedScreens.jsp, Glycan Array: 2505).

### 4.11. Tissue Sections and Immunohistochemistry

Ethanol-fixed human trachea, esophagus, stomach, duodenal junction, jejunum, colon, pancreas, liver, endometrium, ovary, cervix, vagina and oviduct samples were obtained from organ donors. Eighteen tissue samples from 10 subjects were used to prepare a tissue microarray (TMA) of healthy tissues. The samples were obtained from the Nantes University Hospital Center for Biological Resources (http://relib.fr), under the Cancerology Program approved by the Ministry of Research (approval DC-2011-1399).

The tissue sections were deparaffinized and endogenous peroxidases were blocked by incubating the sections with PBS containing 3% hydrogen peroxide (*v/v*) for 5 min. The sections were then blocked with 5% (*w/v*) BSA in PBS for 30 min at room temperature, which was followed by incubation with purified and biotin-labeled human anti-Le^C^ diluted to 300 or 10 μg/mL, for 1 h at 37 °C. After washing twice with PBS, the sections were incubated with peroxidase-labeled avidin (1:1000) and developed using an AEC detection kit (Vector Laboratories) according to the manufacturer’s instructions. The developed slides were washed twice with PBS and counterstained with hematoxylin. After washing with water, the sections were dehydrated, mounted and imaged with a NanoZoomer slide-scanner (Hamamatsu, Hamamatsu City, Japan).

## 5. Conclusions


Human antibodies isolated using disaccharide Le^C^ do not recognize the disaccharide in terminal position of more complex glycans; this paradoxical epitope specificity was not found for similarly isolated murine antibodies. We hypothesize that these immunoglobulins recognize a stable molecular pattern formed by two or more molecules of the tumor (but not normal) cell membrane.Titers of anti-Le^C^ IgM are reduced in patients with breast cancer comparing to healthy donors [4,5]. Together with the cytotoxic effect on breast cancer cells this suggests that anti-Le^C^ have a tumor-surveillance role.Anti-Le^C^ antibodies bind (and presumably are cytotoxic) to cultured breast cancer cells when the monolayer reaches a high density.


## Figures and Tables

**Figure 1 ijms-21-06511-f001:**
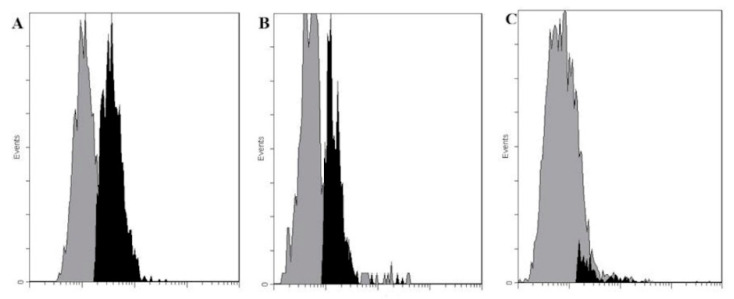
Interaction of human antibodies affinity-isolated using Le^C^-Sepharose with (**A**) HS758t cells, (**B**) ZR 75-1 cells and (**C**) SKBR-3 cells, flow cytometry data (Cytomics FC 500 Beckman Coulter). The concentration of anti-Le^C^ antibodies was 5 µg/mL (1 µg per 10^6^ cells). Cells detached from a monolayer were immediately analyzed using flow cytometry: the gray zone, control (no anti-Le^C^ Abs); the black zone, binding of anti-Le^C^ Abs to cells. The results of flow cytometry data were processed using the CXP Analysis 2.2 software. (**A**) gray zone, mean fluorescence intensity (MFI) 1.2; black zone, MFI 4.3, 66% of positive cells; (**B**) gray zone, MFI 0.53; black zone, MFI 1.4, 68% of positive cells; (**C**) gray zone, MFI 5.39; black zone, MFI 9.05, 7%, that is, practically no positive cells.

**Figure 2 ijms-21-06511-f002:**
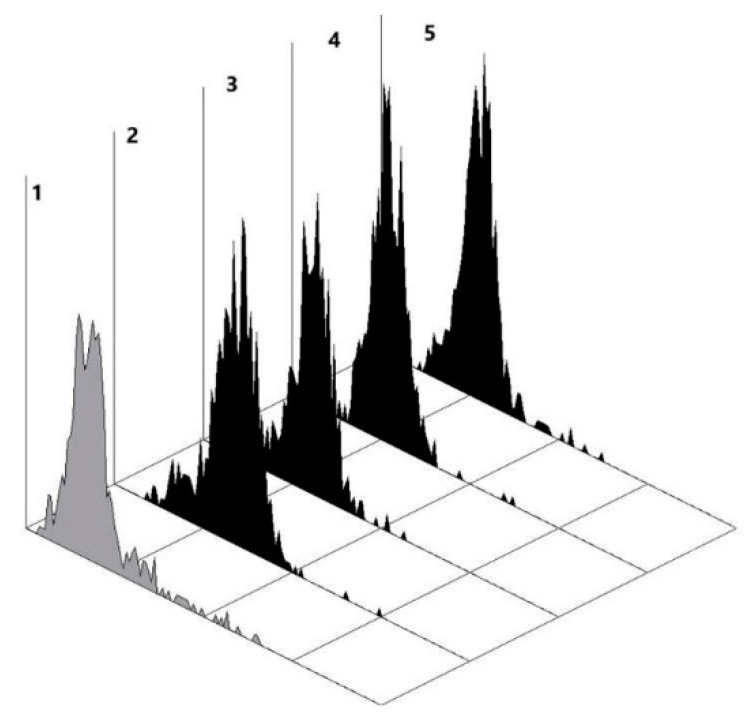
Interaction of human antibodies affinity-isolated using Le^C^-Sepharose with ZR 75-1 cells (at their *max* density), flow cytometry data (Cytomics FC 500 Beckman Coulter). The concentration of anti-Le^C^ antibodies was 5 µg/mL (1 µg per 10^6^ cells). Cells detached from a monolayer were immediately analyzed using flow cytometry: zone **1**, control (no anti-Le^C^ Abs, MFI 0.55); zone **2**, 20 min of incubation with anti-Le^C^ antibodies (95% of positive cells, MFI 1.3); zone **3**, 40 min of incubation with anti-Le^C^ antibodies (95% of positive cells, MFI 1.4); zone **4**, 60 min of incubation with anti-Le^C^ antibodies (96% of positive cells, MFI 1.5); zone **5**, 120 min of incubation with anti-Le^C^ antibodies (87% of positive cells, MFI 2.3).

**Figure 3 ijms-21-06511-f003:**
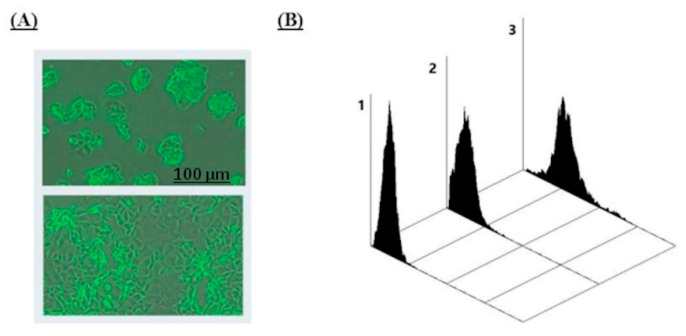
(**A**) Morphology of ZR 75-1 cells grown to 50% (top) and 85% (bottom) monolayer density. (**B**) Interaction of human antibodies affinity-isolated using Le^C^-Sepharose with ZR 75-1 cells, flow cytometry data (Cytomics FC 500 Beckman Coulter). The concentration of anti-Le^C^ antibodies was 5 µg/mL (1 µg per 10^6^ cells). Cells detached from a monolayer were immediately analyzed using flow cytometry: zone **1**, control (no anti-Le^C^ Abs, MFI 0.31); zone **2**, binding of anti-Le^C^ Abs to cells of grown to a density of 50% (7% of cells are positive, MFI 0.34); zone **3**, binding to cells grown to a density of 85% (81% of cells are positive, MFI 2.3).

**Figure 4 ijms-21-06511-f004:**
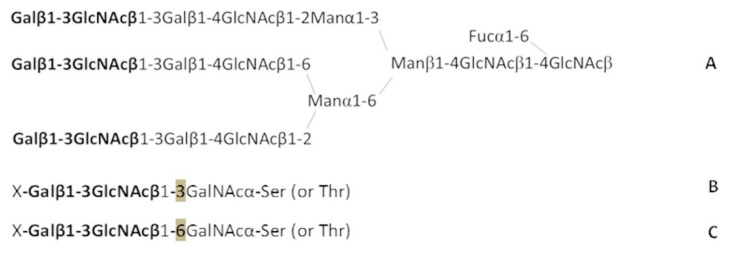
Structures of the discussed N-and O-glycoprotein chains: **A**, triantennary N-chain; **B**, the Le^C^ disaccharide is attached to the 3-position of the inner N-acetylgalactosamine of the O-chain; **C**, the Le^C^ disaccharide is attached to the 6-position of the inner N-acetylgalactosamine of the O-chain. X is a substituent at position 3 or 6 of the Galβ moiety.

**Table 1 ijms-21-06511-t001:** Specificity of mice antibodies (IgG+IgM+IgA) isolated with Le^C^-Sepharose, printed glycan array (PGA) data. The outmost and branch-type Le^C^ motifs are underlined; the innermost ones are shown in grey. Fm, formyl group, that is, –C(O)H.

Nr	Glycan Structure	Median RFU *10^−3^
542	Galβ1-3GlcNAcβ1-3Galβ1-4(Fucα1-3)GlcNAcβ1-6(Galβ1-3GlcNAcβ1-3)Galβ1-4Glcβ	44.1
403	Galβ1-3GlcNAcβ1-3Galβ1-4GlcNAcβ	36.5
376	Galβ1-3GlcNAcβ1-3Galβ1-4Glcβ	35.4
379	Galβ1-3GlcNAcβ1-3Galβ1-4GlcNAcβ	31.8
401	Galβ1-3GlcNAcβ1-3Galβ1-3GlcNAcβ	24.2
538	Galβ1-4(Fucα1-3)GlcNAcβ1-6(Galβ1-3GlcNAcβ1-3)Galβ1-4Glcβ	23.5
399	Galβ1-3GlcNAcα1-3Galβ1-3GlcNAcβ	16.9
529	Neu5Acα2-6(Galβ1-3)GlcNAcβ1-3Galβ1-4Glcβ	16.9
279	Galβ1-3GlcNAcα1-3GalNAcα	10.9
378	Galβ1-3GlcNAcα1-3Galβ1-4GlcNAcβ	10.7
145	Galβ1-3(6-O-Su)GlcNAcβ	7.0
085	Galβ1-3GlcNAcβ (Le^C^ disaccharide)	4.6
144	Galβ1-3(6-O-Su)GlcNAcβ	3.6
397	Galβ1-3GlcN(Fm)β1-3Galβ1-4GlcNAcβ	3.5
381	Galβ1-3GlcNAcβ1-6Galβ1-4GlcNAcβ	3.2

**Table 2 ijms-21-06511-t002:** The frequency of occurrence in the five infants studied of those antibodies (IgM) that showed a high signal level in the cohort of adult donors. Right column: the number of infants (out of a total of five) with significant binding; glycans related to Le^C^ family are shown in bold. Fm, formyl group, i.e., –C(O)H.

Nr	Glycans	Binding of Infant Ig to the Glycan, Number of Positives
020	Rhaα	3/5
080	Galα1-3GlcNAcβ	4/5
243	GlcNAcα1-3Galβ1-4GlcNAcβ	5/5
399	Galβ1-3GlcNAcα1-3Galβ1-3GlcNAcβ	2/5
103	GalNAcα1-3GalNAcα	4/5
102	GalNAcα1-3Galβ	5/5
142	GlcNAcα1-3GalNAcβ	5/5
101	GalNAcα1-3GalNAcβ	5/5
074	Fucβ1-3GlcNAcβ	0/5
118	GlcNAcβ1-6GalNAcα	5/5
331	Neu5Gcα2-3Galβ1-3GlcNAcβ	0/5
307	KDNα2-3Galβ1-3GlcNAcβ	0/5
055	3-O-Su-GlcNAcβ	2/5
267	GlcNAcβ1-3Galβ1-3GlcNAcβ	0/5
264	Galβ1-4Galβ1-4GlcNAcβ	5/5
085	Galβ1-3GlcNAcβ	0/5
401	Galβ1-3GlcNAcβ1-3Galβ1-3GlcNAcβ	0/5
378	Galβ1-3GlcNAcα1-3Galβ1-4GlcNAcβ	3/5
397	Galβ1-3GlcN(Fm)β1-3Galβ1-4GlcNAcβ	0/5
081	Galα1-4GlcNAcβ	3/5
375	Galα1-4GlcNAcβ1-3Galβ1-4GlcNAcβ	4/5
082	Galα1-4GlcNAcβ	0/5
072	Fucα1-3GlcNAcβ	0/5
113	GlcNAcβ1-3GalNAcα	5/5
149	GlcNAcβ1-4(6-O-Su)GlcNAcβ	2/5
117	GlcNAcβ1-4GlcNAcβ-Gly	4/5
382	Galβ1-3GalNAcβ1-4Galβ1-4Glcβ	0/5
164	GlcAβ1-3GlcNAcβ	0/5
380	Galβ1-3GlcNAcα1-6Galβ1-4GlcNAcβ	0/5
161	6-O-Su-Galβ1-3GlcNAcβ	0/5
073	Fucα1-4GlcNAcβ	0/5
299	Neu5Acα2-3Galβ1-3GlcNAcβ	0/5

**Table 3 ijms-21-06511-t003:** Profiling of human antibodies (IgM) isolated from immunoglobulin preparation (CIP) using the Le^C^-Sepharose affinity adsorbent. Only top-rank glycans are shown. The outmost Le^C^ motifs are underlined; the innermost ones are shown in grey. Only relative fluorescence units (RFU) values >5000 are shown. Fm, formyl group, that is, –C(O)H.

Nr	Structure	Median RFU*10^−3^
398	Galβ1-3GlcN(Fm)β1-3Galβ1-3GlcNAcβ	35.1
399	Galβ1-3GlcNAcα1-3Galβ1-3GlcNAcβ	32.6
331	Neu5Gcα2-3Galβ1-3GlcNAcβ	28.3
080	Galα1-3GlcNAcβ	26.4
401	Galβ1-3GlcNAcβ1-3Galβ1-3GlcNAcβ	24.4
085	Galβ1-3GlcNAcβ (Le^C^ disaccharide)	23.8
074	Fucβ1-3GlcNAcβ	23.2
307	KDNα2-3Galβ1-3GlcNAcβ	22.9
299	Neu5Acα2-3Galβ1-3GlcNAcβ	22.9
267	GlcNAcβ1-3Galβ1-3GlcNAcβ	22.5
154	3-O-Su-Galβ1-3GlcNAcβ	22.5
055	3-O-Su-GlcNAcβ	17.7
164	GlcAβ1-3GlcNAcβ	13.8
161	6-O-Su-Galβ1-3GlcNAcβ	11.0
130	6-O-Bn-Galβ1-3GlcNAcβ	10.2
010	GlcNAcβ	8.3
072	Fucα1-3GlcNAcβ	6.2
378	Galβ1-3GlcNAcα1-3Galβ1-4GlcNAcβ	5.6
397	Galβ1-3GlcN(Fm)β1-3Galβ1-4GlcNAcβ	5.2

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
