# Peer review of "Human Natural Antibodies Recognizing Glycan Galβ1-3GlcNAc (LeC)"

_ijms, 2020, doi:10.3390/ijms21186511_

Round 1

Reviewer 1 Report

The authors present an interesting study on the study of the epitope specificity of IgM antibodies raised against LeC versus a variety of glycans. The study has been performed in a very professional manner. A printed glycan array has been employed to verify the binding, using an ELISA method. The avidity for a multivalent molecule was estimated in the nano molar range, which makes it feasible to be used for physiological testing. Different breast cancer cell lines have also been employed, showing certain discrimination. Generally speaking, the work is interesting, technically sound and has results of interest for the scientific community.

Author Response

Point 1: The authors present an interesting study on the study of the epitope specificity of IgM antibodies raised against LeC versus a variety of glycans. The study has been performed in a very professional manner. A printed glycan array has been employed to verify the binding, using an ELISA method. The avidity for a multivalent molecule was estimated in the nano molar range, which makes it feasible to be used for physiological testing. Different breast cancer cell lines have also been employed, showing certain discrimination. Generally speaking, the work is interesting, technically sound and has results of interest for the scientific community.

Response 1: We appreciate your positive comment.

Reviewer 2 Report

The work described by Dobrochaeva et al. aimed at characterizing the epitope specificity of natural antibodies in human serum to a specific carbohydrate antigen Galβ1-3GlcNAc (designated LeC). This work reflects the efforts of the authors to identify the molecular targets and relevance of these naturally occurring antibodies, as there is evidence that these antibodies may have a role in tumour immune-surveillance. The work described here has novelty and significance and is of interest for the readership of IJMS.

The authors state that human antibodies exhibit ‘unusual’ binding specificity not recognizing the Lec disaccharide when presented as a terminal part of a carbohydrate chain in glycoproteins, as far characterized. To refine the specificity of the antibodies the authors apply a diversified glycan microarray with synthetic oligosaccharides containing Galβ1-3GlcNAcβ sequences. The authors also analyse the binding of the antibodies isolated to human healthy and colorectal carcinoma tissues and tumour cell lines. The research design is appropriate, but, in this reviewer opinion, for publication the manuscript should be revised to present the results clearly and support author’s conclusions.

Major points

  1. The major results to support glycan binding properties of the antibodies (from mouse or human serum, affinity purified or in whole serum) at the molecular level are derived from glycan microarrays, and different glycan microarray platforms have been used. More information on the different types of arrays used e.g. their quality control is required. Or if the microarrays were published, a clearly reference to the publication and what type of quality control was used there. It is not clear if the same arrays were used for results presented in Tables 1-3. The authors should consider the MIRAGE guidelines for reporting glycan microarray data and refer these.
  2. Description and presentation of glycan microarray data – the results are tabulated (Tables 1-3), listing the top ranking glycans and in supplementary material. This is fine as the fluorescence intensity values are provided. However, it will be difficult for the general reader to follow with the supplementary data and relate to the structures in the different analysis. A possible way to improve would be to refer to the ID numbers of the probes.  
  3. Results 2.1 line 79- Not clear what the authors meant by ‘Similarly to human antibodies, the affinity of mouse anti-LeC for LeC disaccharide is moderate’. What is the measurement of affinity here? When comparing the mouse serum with human serum has the sum IgG + IgM antibodies been considered for the human antibodies? Where is the data?
  4. For the results of tissue and cell staining with human anti-LeC antibodies, could the authors comment on the relevance of the detection of whole Igs vs IgM detection? Is any difference anticipated? Were mouse antibodies tested? It would be of interest to compare.
  5. Results 2.7 - Binding in presence of Galβ1-3GlcNAc disaccharide. The authors should describe in more detail the spacer effects on binding. Was the disaccharide presented with different spacers in microarrays or ELISA?
  6. Discussion, lines 270-272- the conclusion here is misleading and should be revised.
  7. In the Discussion the authors highlight the N-glycan structure with terminal Galβ1-3GlcNAc that is not bound by the anti-Lec human antibodies. A suggestion for reader’s comprehensiveness and easily grasping author’s conclusions is to highlight also the structures discussed that could be part of the recognized epitope e.g. the O-glycan backbone the authors predict.
  8. In the reviewer understanding, the last 2 conclusion points (lines 489-493) are not supported by the results presented in the current work and should be revised.

Other points

  1. Introduction lines 60-62. Two monoclonal antibodies LU-BCRU-G7 and 58-1 are introduced. The information is not clear or enough in what their glycan binding specificities are, if they are known. The authors do not refer further to these antibodies or correlate the specificities with their findings. Could the authors comment here?
  2. Introduce Bar scale in Figure 3A

Author Response

The work described by Dobrochaeva et al. aimed at characterizing the epitope specificity of natural antibodies in human serum to a specific carbohydrate antigen Galβ1-3GlcNAc (designated LeC). This work reflects the efforts of the authors to identify the molecular targets and relevance of these naturally occurring antibodies, as there is evidence that these antibodies may have a role in tumour immune-surveillance. The work described here has novelty and significance and is of interest for the readership of IJMS.

The authors state that human antibodies exhibit ‘unusual’ binding specificity not recognizing the Lec disaccharide when presented as a terminal part of a carbohydrate chain in glycoproteins, as far characterized. To refine the specificity of the antibodies the authors apply a diversified glycan microarray with synthetic oligosaccharides containing Galβ1-3GlcNAcβ sequences. The authors also analyse the binding of the antibodies isolated to human healthy and colorectal carcinoma tissues and tumour cell lines. The research design is appropriate, but, in this reviewer opinion, for publication the manuscript should be revised to present the results clearly and support author’s conclusions.

Major points

Point 1: The major results to support glycan binding properties of the antibodies (from mouse or human serum, affinity purified or in whole serum) at the molecular level are derived from glycan microarrays, and different glycan microarray platforms have been used. More information on the different types of arrays used e.g. their quality control is required. Or if the microarrays were published, a clearly reference to the publication and what type of quality control was used there. It is not clear if the same arrays were used for results presented in Tables 1-3. The authors should consider the MIRAGE guidelines for reporting glycan microarray data and refer these. 

Response 1: The corresponding information was added to the Supplementary data.

Point 2: Description and presentation of glycan microarray data – the results are tabulated (Tables 1-3), listing the top ranking glycans and in supplementary material. This is fine as the fluorescence intensity values are provided. However, it will be difficult for the general reader to follow with the supplementary data and relate to the structures in the different analysis. A possible way to improve would be to refer to the ID numbers of the probes.

Response 2: Thanks for the advice. It was corrected in the manuscript.

Point 3: Results 2.1 line 79- Not clear what the authors meant by ‘Similarly to human antibodies, the affinity of mouse anti-LeC for LeC disaccharide is moderate’. What is the measurement of affinity here? When comparing the mouse serum with human serum has the sum IgG + IgM antibodies been considered for the human antibodies? Where is the data?

Response 3: We meant that the rank of antibodies in comparison with other PGA glycans was moderate. Current text is: Similarly to human antibodies, the affinity of mouse anti-LeC for LeC disaccharide is moderate (Table 1, where it is not at the top position), moreover, the difference (measured in relative fluorescence units, RFU) between the affinities to LeC disaccharide and the top glycans is as high as an order of magnitude. We change this phrase as follows: "We can indirectly assume that similarly to human antibodies, the affinity of mouse anti-LeC for LeC disaccharide is moderate, because this glycan occupies not the top position, moreover, the difference (measured in relative fluorescence units, RFU) between  LeC disaccharide and the top glycans is as high as an order of magnitude."

Point 4: For the results of tissue and cell staining with human anti-LeC antibodies, could the authors comment on the relevance of the detection of whole Igs vs IgM detection? Is any difference anticipated? Were mouse antibodies tested? It would be of interest to compare.

Response 4: In these experiments, we used only antibodies that were a sum of IgM and IgG, we wanted to be as close as possible to the "natural" situation that takes place in vivo. Although, given the strong prevalence of IgM, it can be assumed in a first approximation that the presented results are more likely to be related to IgM. Thanks for the idea, we will try to do immunostaining with the help of the IgG fraction in the future. We did not use mouse antibodies to study interactions with cells and tissues.

Point 5: Results 2.7 - Binding in presence of Galβ1-3GlcNAc disaccharide. The authors should describe in more detail the spacer effects on binding. Was the disaccharide presented with different spacers in microarrays or ELISA?

Response 5: In the experiments on inhibition (2.7), we used only a disaccharide with a C3 spacer. Because in previous publications and in the article in press [18], we have demonstrated only a minor effect of the nature of the spacer on the binding of antibodies in ELISA and PGA. We will add the corresponding phrase to (2.7): "In previous publications [1, 18], we demonstrated only minor effect of the nature of the spacer on the binding of antibodies in ELISA and PGA, so here we took only a disaccharide with a C3 spacer."

Point 6: Discussion, lines 270-272- the conclusion here is misleading and should be revised.

Response 6: We are agree, new version is: "From the data presented, a paradoxical conclusion could be drawn that 4OH and 5CH2OH of GlcNAc form a key epitope. However, the data on the absence (or weak) binding of the antibodies to many of other PGA glycans containing the same combination of 4OH and 5CH2OHgroups suggest the opposite. This again points to the unusual "architecture" of the anti-LeC antibody epitope."

Point 7: In the Discussion the authors highlight the N-glycan structure with terminal Galβ1-3GlcNAc that is not bound by the anti-LeC human antibodies. A suggestion for reader’s comprehensiveness and easily grasping author’s conclusions is to highlight also the structures discussed that could be part of the recognized epitope e.g. the O-glycan backbone the authors predict.

Response 7: Thanks for the advice. We have placed a new one, Fig. 4, which includes all the discussed structures of fragments of the O-chain, and added the corresponding numbers in the body of the Discussion.

Point 8: In the reviewer understanding, the last 2 conclusion points (lines 489-493) are not supported by the results presented in the current work and should be revised.

Response 8:

  • Titers of anti-LeC IgM are reduced in patients with breast cancer comparing to healthy donors. Together with the cytotoxic effect on breast cancer cells this suggests that anti-LeC have a tumor-surveillance role.
  • Anti-LeC antibodies are cytotoxic to breast cancer cell culture if the monolayer reaches a high density.

In the first conclusion, we combine data from this manuscript with data published earlier. It is especially important for us to summarize all the currently known results related to the antibodies. The revised version adds the corresponding refs [4,5].

Indeed, the second conclusion is incorrect, we got ahead of ourselves here - the cytotoxic effect depending on the cell density will be discussed in our next publication, whereas here we describe (see lines 176-177) only binding. The revised version would be as follows: "Anti-LeC antibodies bind (and presumably are cytotoxic) to cultured breast cancer cells when the monolayer reaches a high density."

Other points

Point 1: Introduction lines 60-62. Two monoclonal antibodies LU-BCRU-G7 and 58-1 are introduced. The information is not clear or enough in what their glycan binding specificities are, if they are known. The authors do not refer further to these antibodies or correlate the specificities with their findings. Could the authors comment here?

Response 1: The specificity and (in short) comparison with human antibodies are presented in our article [10], we did it exactly so as not to overload the text. We now add a footnote: "1Specificity and comparison of mAb with human anti-LeC are presented in [10]."

Point 2: Introduce Bar scale in Figure 3A.

Response 2: The Figure 3A was corrected.

Round 2

Reviewer 2 Report

The authors adequately answered major points raised, and the manuscript may be accepted in the present form.